# Colistin Monotherapy versus Colistin plus Meropenem Combination Therapy for the Treatment of Multidrug-Resistant *Acinetobacter baumannii* Infection: A Meta-Analysis

**DOI:** 10.3390/jcm11113239

**Published:** 2022-06-06

**Authors:** Chienhsiu Huang, Ihung Chen, Tiju Tang

**Affiliations:** Department of Internal Medicine, Dalin Tzu Chi Hospital, Buddhist Tzu Chi Medical Foundation, Hualien 97002, Taiwan; b89401098@nut.edu.tw (I.C.); df139191@tzuchi.com.tw (T.T.)

**Keywords:** colistin, meropenem, combination therapy, *Acinetobacter baumannii*, multidrug-resistant

## Abstract

(1) Introduction: Colistin combination therapy with other antibiotics is a way to enhance colistin activity. The purpose of this meta-analysis was to compare the efficacy and safety of treatment with colistin monotherapy versus colistin plus meropenem combination therapy in patients with drug-resistant *Acinetobacter baumannii* infection. (2) Methods: All studies were included if they reported one or more of the following outcomes: clinical improvement, complete microbiological response, 14-day mortality, hospital mortality, or nephrotoxicity. (3) Results: Three randomized controlled trials and seven retrospective studies were included in the meta-analysis. Colistin monotherapy has similar rates of clinical improvement, 14-day mortality, hospital mortality, and nephrotoxicity as colistin plus meropenem combination therapy. Regarding complete microbiological response, the colistin plus meropenem combination was better than colistin monotherapy. (4) Discussion: Previous meta-analyses demonstrated heterogeneity in study quality and a lack of evidence supporting the use of colistin-based combination therapy. Our meta-analysis clearly showed that colistin combined with meropenem was not superior to colistin monotherapy for the treatment of *Acinetobacter baumannii* infection. (5) Conclusions: The efficacy and safety of treatment with colistin monotherapy and that of colistin plus meropenem combination therapy in patients with drug-resistant *Acinetobacter baumannii* infection were comparable. The majority of the evidence was obtained from nonrandomized studies, and high-quality randomized controlled trials are needed to confirm the role of colistin plus meropenem combination therapy in the treatment of multidrug-resistant *Acinetobacter baumannii* infection.

## 1. Introduction

*Acinetobacter baumannii* is an aerobic Gram-negative opportunistic bacterium that is an important pathogen of nosocomial infections worldwide. Carbapenems have been considered to be appropriate agents to treat *Acinetobacter baumannii* infection. *Acinetobacter baumannii* has the ability to acquire resistance to multiple classes of antimicrobial agents, and a worldwide surge in carbapenem resistance has been reported [1,2]. The presence of multidrug-resistant (MDR) *Acinetobacter baumannii* has increased the prevalence of *Acinetobacter baumannii* infection. MDR *Acinetobacter baumannii*, which is resistant to all standard antimicrobial agents, makes the choice of appropriate antimicrobial treatment difficult. Colistin is used extensively to treat MDR *Acinetobacter baumannii* infection and remains an important active antibiotic to treat MDR *Acinetobacter baumannii* [3,4,5]. The lipopolysaccharide present at the surface of the outer membrane of Gram-negative bacteria (GNB) prevents the penetration of hydrophobic and/or large antibiotics. The outer membrane of GNB is the target for colistin action. Colistin has both hydrophilic (https://en.wikipedia.org/wiki/Hydrophile, accessed on 27 May 2022) and lipophilic moieties (https://en.wikipedia.org/wiki/Lipophilicity, accessed on 27 May 2022). The cationic region interacts with the negatively charged outer membrane of GNB and competitively displaces calcium (Ca^2+^) and magnesium (Mg^2+^) ions from the phosphate groups found in membrane lipids. Therefore, the lipopolysaccharide becomes destabilized, which consequently increases the permeability of the bacterial membrane and leads to leakage of the cytoplasmic content and the disruption of the outer membrane [6] However, emerging colistin-resistant *Acinetobacter baumannii* is a complex and important issue. In a study by Nowak et al., 45.5% of 65 clinically isolated *Acinetobacter baumannii* isolates recovered from respiratory tract samples from patients with pneumonia were resistant to colistin [7]. There are several ways to enhance colistin activity, including an increased loading dose, a higher maintenance dose, and combination therapy with other antibiotics. Regarding combination therapy, the use of a carbapenem combination with colistin was able to achieve synergistic killing of *Acinetobacter baumannii*. For example, the synergistic killing of *Acinetobacter baumannii* was recently demonstrated in many in vitro studies. Various colistin combinations have been explored, including those containing carbapenems, tigecycline, sulbactam, aminoglycosides, rifampicin, and fosfomycin. The mechanisms underlying the synergistic activity achieved by these combinations are not fully understood. This effect is considered to be mediated by the permeabilizing effect of colistin on the bacterial outer membrane, which permits the entry of large hydrophobic molecules. This disruption of the membrane may have a positive impact on the functions of several antibiotics and may result in improvements in their activities [8,9,10,11]. In vitro synergy studies are limited in their ability to predict clinical effects. Colistin has increasingly been used in combination with other antibiotics for the treatment of MDR *Acinetobacter baumannii* infection [12]. The use of colistin-based combination therapy may prevent emerging resistance and preserve the activity of colistin against *Acinetobacter baumannii* [13]. Clinical studies did not show consistent results regarding the effect of colistin-based combination therapy in MDR *Acinetobacter baumannii* infection patients. The heterogeneous combinations of antibacterial agents included carbapenems, sulbactam, tigecycline, rifampicin, fosfomycin, aminoglycosides, and vancomycin. In 2018, Papst et al. conducted an international cross-sectional, internet-based questionnaire survey to explore the contemporary antibiotic management of infections caused by carbapenem-resistant GNB in hospitals. The study showed that combination therapy was occasionally prescribed in hospitals. Respondents were likely to consider combination therapy for infections caused by *Enterobacteriaceae*, *Pseudomonas aeruginosa*, and *Acinetobacter baumannii*. The combination of a polymyxin with a carbapenem was used in most cases. The author concluded that combination therapy was the preferred treatment strategy for infections caused by carbapenem-resistant GNB among hospital representatives even though high-quality evidence of the efficacy of carbapenem-based combination therapy is lacking [14]. Whether colistin used in combination with other antibiotics results in enhanced activity against drug-resistant *Acinetobacter baumannii* and whether this leads to improved clinical outcomes are unclear. Which combination regimens are safe and effective in clinical settings is still controversial, and the optimal agent for combinations with colistin for the treatment of *MDR Acinetobacter baumannii* infection is undefined. Previous meta-analyses evaluated the efficacy and safety of colistin as a monotherapy or in combination with other antibacterial agents to treat MDR GNB infections. Carbapenems combined with polymyxins had better synergy in their effect on *Acinetobacter baumannii* than on other bacteria and better synergy with meropenem than with imipenem [8]. In this meta-analysis, we compared the clinical outcomes of colistin monotherapy versus colistin in combination with meropenem. The purpose of this meta-analysis was to compare the efficacy and safety of treatment with intravenous colistin monotherapy and intravenous colistin combined with meropenem in patients with MDR *Acinetobacter baumannii* infection.

## 2. Methods

### 2.1. Data Search

All clinical studies were identified by a comprehensive literature search in the PubMed, Web of Science, and Cochrane Library databases between 1 January 2000 and 28 February 2022, and those that examined treatment options, including colistin monotherapy and colistin combined with another antibacterial agent for the treatment of *Acinetobacter baumannii* infection, were included. The search terms included “colistin or polymyxin”, “Acinetobacter baumannii”, and “combination therapy” and were used to identify relevant clinical studies, including randomized controlled trials (RCTs) and retrospective and prospective cohort studies. Articles containing relevant terms in each database were identified and imported into Endnote Library for the deletion of duplicate records. After excluding duplicates, all studies were reviewed by reading the title and/or abstract to identify irrelevant studies. To determine the eligibility of the identified trial reports, two of the authors independently screened the titles and abstracts. After excluding irrelevant studies, all of the relevant articles were reviewed by reading the full text to determine eligible trial reports.

### 2.2. Inclusion and Exclusion Criteria

We included the findings of observational studies to help offset the limitations of data analysis because a limited number of RCTs were available. The studies were considered eligible for inclusion only if they directly compared the clinical effectiveness of colistin monotherapy and colistin-based combination therapies involving meropenem in the treatment of adult patients with *Acinetobacter baumannii* infection. Data were manually extracted from the eligible full-text articles. Author, year of publication, region, study type, infection site, resistance pattern of *Acinetobacter baumannii*, total number of monotherapy group patients, total number of combination group patients, and antibiotic dosage were extracted. Colistin was administered as a 9 million units (MU) loading dose, followed by 4.5 MU maintenance doses administered every 12 h, adjusted for renal function in patients with creatinine clearance. A loading dose for colistin that was not recommended in the studies was included also. Meropenem was given as 1–2 gm every 8 h, adjusted for renal function. The duration of antibiotic treatment was not limited in this study. All studies were included if they reported one or more of the following outcomes: clinical improvement, complete microbiological response, 14-day mortality, hospital mortality or 28-day mortality, or nephrotoxicity.

We did not include studies that evaluated inhaled colistin therapy or studies that focused on in vitro activity or pharmacokinetic-pharmacodynamic assessment. We excluded studies that were primarily reviews, meta-analyses, guidelines, case reports, editorials, and animal studies, and those in which treatment regimens that did not include carbapenem and colistin. Studies with a population <18 years old were excluded. Studies reporting no more than five patients per treatment group were excluded. Articles published in all languages were included.

MDR refers to nonsusceptibility to ≥1 agent in ≥3 antimicrobial categories. Extensive drug resistance (XDR) refers to nonsusceptibility to ≥1 agent in all but ≤2 antimicrobial categories. Nonsusceptibility to all antimicrobial agents tested was defined as pandrug resistance (PDR) [15].

### 2.3. Quality Assessment

We assessed the risk of bias in each study using the Cochrane Risk-of-Bias Tool 2.0 for RCTs. The Risk of Bias in Non-randomized Studies of Interventions (ROBINS-I) tool was used to evaluate observational studies [16]. We conducted a sensitivity analysis by systematically removing each study and assessing the impact of the study quality on the effect estimates. Quality of the evidence was ranked based on the risk of bias according to the Grading of Recommendations Assessment, Development and Evaluation (GRADE) approach at the outcome level [17,18]. Two reviewers examined publications independently to avoid bias. When disagreement occurred, a third author resolved the issue.

### 2.4. Statistical Analysis

Data were entered into the Cochrane Review Manager software RevMan 5. Differences were expressed as odds ratios (ORs) with 95% confidence intervals (CIs) for dichotomous outcomes. The significance of the pooled ratios was determined by the Z test, and a *p* value less than 0.05 was considered statistically significant. The I^2^ test was used to assess the proportion of statistical heterogeneity, and the Q-statistic test was used to define the degree of heterogeneity. A *p* value less than 0.10 for the Q-test and I^2^ more than 50% were considered significant among the studies. The fixed-effects model was used when the effects were assumed to be homogenous, while the random-effects model was used when they were heterogeneous. Publication bias was assessed by examining the funnel plot.

## 3. Results

### 3.1. Characteristics of the Included Trials

The numbers of initial search results from PubMed, Web of Science, and the Cochrane Library were 417, 468, and 80, respectively. There were 487 duplicate articles. A total of 404 irrelevant studies were identified by reading the title and/or abstract. After excluding duplicates and irrelevant studies, 74 potentially relevant articles remained. After full-text article review, 60 articles were excluded because they lacked results of comparisons between colistin monotherapy and combination therapy with colistin and meropenem in patients with *Acinetobacter baumannii* infection. Three articles were excluded because the monotherapy group and/or combination therapy group did not include more than five patients per treatment group [19,20,21]. Another article was excluded because there were no detailed patient numbers for the monotherapy group and combination therapy group [22]. Finally, 10 studies were included in the meta-analysis [23,24,25,26,27,28,29,30,31,32]. The main characteristics of the 10 included studies are shown in Table 1 and Figure 1. Of them, three were from Korea; three were from Israel; two were from Turkey; one was from Greece; and one was from Thailand. Three were RCTs, and seven were retrospective observational studies. Eight studies had high risk of bias. The risk of bias assessment is presented in Figure 2 and Table 2.

### 3.2. Efficacy and Safety Outcomes

Nine studies involving 1484 patients (660 with colistin monotherapy, 824 with colistin plus meropenem combination therapy) reported clinical improvement. There was no statistically significant difference in clinical improvement between patients treated with colistin monotherapy and colistin plus meropenem combination therapy (OR = 0.85, 95% CI = 0.67–1.08, *p* = 0.19, I^2^ = 7%) (Figure 3).

Five studies involving 655 patients (285 with colistin monotherapy, 370 with colistin plus meropenem combination therapy) reported the microbiological response. The overall microbiological response was significantly different between the two groups (OR = 0.71, 95% CI = 0.51–0.98, *p* = 0.04, I^2^ = 36%). The combination therapy patient group had a better microbiological response than the monotherapy patient group (Figure 4).

Five studies involving 949 patients (441 with colistin monotherapy, 508 with colistin plus meropenem combination therapy) reported 14-day mortality. There was no statistically significant difference in 14-day mortality between patients treated with colistin monotherapy and colistin plus meropenem combination therapy (OR = 1.04, 95% CI = 0.79–1.37, *p* = 0.77, I^2^ = 37%) (Figure 5).

Seven studies involving 1269 patients (596 with colistin monotherapy, 673 with colistin plus meropenem combination therapy) reported hospital mortality or 28-day mortality. There was no statistically significant difference in hospital mortality or 28-day mortality between patients treated with colistin monotherapy and colistin plus meropenem combination therapy (OR = 0.94, 95% CI = 0.75–1.17, *p* = 0.59, I^2^ = 59%) (Figure 6).

Nephrotoxicity is the main adverse effect of colistin treatment. Data regarding nephrotoxicity were reported in four studies including 535 patients (235 with colistin monotherapy, 300 with colistin plus meropenem combination therapy). There was no statistically significant difference in nephrotoxicity between patients treated with colistin monotherapy and colistin plus meropenem combination therapy (OR = 1.34, 95% CI = 0.92–1.95, *p* = 0.13, I^2^ = 0%) (Figure 7).

## 4. Discussion

The current meta-analysis of 10 studies (3 RCTs and 7 retrospective observational studies) provides evidence that colistin monotherapy is as effective as colistin plus meropenem combination therapy for the treatment of MDR *Acinetobacter baumannii*, XDR *Acinetobacter baumannii*, and carbapenem resistant (CR) *Acinetobacter baumannii* infections. In terms of complete microbiological response, the colistin plus meropenem combination patient group showed better results than the colistin monotherapy patient group. The microbiological response was the only statistically significant difference between the two groups.

There were eight meta-analyses reported in the literature which explored the issue of intravenous colistin versus colistin-based combination therapy against multidrug-resistant GNB infections (Table 3). In the meta-analysis by Chen et al. (2015), no difference in clinical improvement, hospital mortality, or nephrotoxicity was found between the colistin monotherapy and colistin combined with other antibiotics groups for the treatment of drug-resistant *Acinetobacter baumannii* infection. However, colistin-based combination therapy was shown to increase the microbiological response [33]. In the meta-analysis performed by Zusman et al. (2017), no difference in all-cause mortality was found between patients treated with colistin monotherapy and those treated with colistin-based combination therapy with other antibiotics against carbapenem-resistant GNB infections (*Acinetobacter baumannii, Klebsiella pneumoniae*, and *Pseudomonas aeruginosa*). [34]. In the meta-analysis by Kengkla et al. (2018), there was no statistically significant difference in clinical cure outcomes among colistin-based combinations with other antibiotic therapies in patients with MDR *Acinetobacter baumannii* and XDR *Acinetobacter baumannii* infections. However, colistin in combination with sulbactam was associated with a significantly higher microbiological cure rate than colistin monotherapy [35]. In the meta-analysis by Vardakas et al. (2018), colistin-containing combination regimens did not decrease mortality in patients with MDR GNB infections (*Acinetobacter baumannii, Klebsiella pneumoniae*, and *Pseudomonas aeruginosa*) [36]. In the meta-analysis by Cheng et al. (2018), which analyzed five RCTs, no differences in all-cause mortality or microbiologic response were found between colistin monotherapy and colistin-based combination therapy with other antibiotics against carbapenem-resistant GNB infections (*Acinetobacter baumannii,*
*Enterobacteriaceae*, and *Pseudomonas aeruginosa*), especially for *Acinetobacter baumannii* infection [37]. In the meta-analysis performed by Wang et al. (2019), the subgroup analysis of the effect of colistin-based combination with carbapenem and/or sulbactam on *Acinetobacter baumannii* infection showed no statistically significant difference in clinical response or 28-day mortality from those achieved with colistin monotherapy, and only exhibited a microbiological benefit [38]. In the meta-analysis performed by Schmid et al. (2019), the study explored monotherapy versus combination therapy for MDR GNB infections (*Acinetobacter baumannii, Klebsiella pneumoniae*, and *Pseudomonas aeruginosa*). The subgroup analysis showed no differences in the clinical cure rates and mortality rate between patients treated with monotherapy and those treated with combination therapy with other antibiotics targeting MDR *Acinetobacter baumannii*/XRD *Acinetobacter baumannii* [39]. The meta-analysis by Samal et al. (2021) explored polymyxin combination therapy with other antibiotics for the treatment of multidrug-resistant GNB infections, including *Acinetobacter baumannii, Klebsiella pneumoniae*, and *Pseudomonas aeruginosa*. The studies revealed no statistically significant difference in mortality between the two groups. However, there was a trend toward mortality benefits with combination therapy containing carbapenem [40].

Regarding clinical improvement, four of the eight meta-analyses reported no statistically significant difference between the effects of colistin monotherapy and colistin combination therapy again *Acinetobacter baumannii* infection [33,35,38,39]. Four other meta-analyses did not report clinical improvement between the two groups. The results of the analysis of clinical improvement in the current meta-analysis were consistent with the results of previous meta-analyses.

Regarding microbiological response, two of the eight meta-analyses reported a statistically significant difference between patients administered colistin monotherapy and those treated with colistin combination therapy targeting *Acinetobacter baumannii* [33,38]. One meta-analysis reported that colistin in combination with sulbactam was associated with a significantly higher microbiological response than that achieved with colistin monotherapy targeting *Acinetobacter baumannii* [35]. One meta-analysis reported no statistically significant difference between patients treated with colistin monotherapy and those administered colistin combination therapy targeting GNB [37]. Three other meta-analyses did not report microbiological response in the two groups. The results of previous meta-analyses showed a better microbiological response in patients receiving combination therapy, and the benefit was observed only in patients treated with combination therapy targeting *Acinetobacter baumannii* infection and not in those treated with therapies targeting all of the GNB infections that were assessed. The results of the analysis of microbiological response were consistent between the previous and the current meta-analyses.

Regarding hospital mortality (or 28-day mortality), all of the eight meta-analyses reported no statistically significant difference between patients treated with colistin monotherapy and those administered colistin combination therapy targeting GNB infections. The results of the analysis of hospital mortality in the current meta-analysis were consistent with the results of previous meta-analyses.

Regarding nephrotoxicity, three of the eight meta-analyses reported no statistically significant difference between patients treated with colistin monotherapy and those receiving colistin combination therapy [35,37,38]. Five other meta-analyses did not report nephrotoxicity in the two groups. The results of the analysis of nephrotoxicity in the current meta-analysis were consistent with the results of previous meta-analyses.

In the eight meta-analyses, four meta-analyses focused on colistin versus colistin in combination with other antibiotics to treat *Acinetobacter baumannii* infection [33,35,38,39]. Three of the four meta-analyses showed that colistin-based combination therapy increased the microbiological response. All four meta-analyses demonstrated that no differences were found between colistin monotherapy and colistin combination therapy for the treatment of *Acinetobacter baumannii* infection in clinical improvement and hospital mortality. The findings were the same as those of our current meta-analysis. 

Previous meta-analyses focused on colistin monotherapy versus colistin-based combination therapy with various antibacterial agents (such as carbapenems, tigecycline, sulbactam, aminoglycosides, fosfomycin, and rifampicin) to treat drug-resistant GNB infections (*Acinetobacter baumannii, Klebsiella pneumoniae, Pseudomonas aeruginosa*, and *Escherichia coli*). Previous meta-analyses included heterogeneous populations with different GNB infections and nonstandardized administration of heterogeneous combination treatment regimens. Therefore, these meta-analyses demonstrated heterogeneity in their study quality, variable study results, and lack of evidence supporting the superiority of colistin-based combination therapy. Our meta-analysis focused on colistin monotherapy versus colistin in combination with meropenem to treat drug-resistant *Acinetobacter baumannii* infection. This meta-analysis also included three newly published studies that were not included in the previous meta-analyses [30,31,32]. Our meta-analysis clearly showed that colistin combined with meropenem was not superior to colistin monotherapy to treat *Acinetobacter baumannii* infection except with regard to microbiological response.

There were three RCTs in our meta-analysis. Paul et al. showed that colistin plus carbapenem combination therapy was not superior to colistin monotherapy in the treatment of carbapenem-resistant *Acinetobacter baumannii* infection. There were no statistically significant differences in clinical improvement, 14-day mortality, or 28-day mortality between the colistin monotherapy patient group and the colistin plus meropenem combination therapy patient group [26]. Dickstein et al. showed no statistically significant difference in 28-day mortality between those who received colistin monotherapy and colistin plus meropenem combination therapy in the treatment of carbapenem-resistant *Acinetobacter baumannii* infection [29]. In the study by Nutman et al., colistin plus meropenem combination therapy was not associated with significant differences in clinical improvement or 14-day mortality compared to monotherapy with colistin in the treatment of carbapenem-resistant *Acinetobacter baumannii* infection [30]. We summarized the results of three RCTs in which treatment with colistin in combination with meropenem did not lead to better clinical improvement or lower mortality than colistin monotherapy in infections caused by carbapenem-resistant *Acinetobacter baumannii*.

## 5. Limitations

This meta-analysis has major strengths. The combination therapy antibiotic was meropenem, and the infectious pathogen included only Acinetobacter baumannii. Only one combination antibiotic and only one infectious pathogen were included in this meta-analysis. Thus, we can make a definite conclusion that colistin plus meropenem combination therapy for the treatment of Acinetobacter baumannii infection is not superior to colistin monotherapy. It is difficult to perform prospective RCTs to explore this issue. The majority of studies incorporated in this meta-analysis were retrospective observational studies, and only three RCTs were included in this meta-analysis. Selection bias and confounding were impossible to eliminate owing to the inherent nature of nonrandomized studies. Thus, our findings should be interpreted with caution. However, the conclusions of this meta-analysis are similar to the conclusions of the three RCTs and previous meta-analyses in the literature. In addition, the number of included studies for a certain comparison was small, which was a limitation of this meta-analysis.

## 6. Conclusions

The current meta-analysis found that colistin monotherapy was associated with similar rates of clinical improvement, 14-day mortality, hospital mortality, and nephrotoxicity to colistin plus meropenem combination therapy. There was low quality evidence that colistin plus meropenem combination therapy demonstrated a microbiological benefit. Therefore, colistin plus meropenem combination therapy was not superior to colistin monotherapy in the treatment of drug-resistant *Acinetobacter*
*baumannii* infection. However, the majority of evidence was from nonrandomized studies, and high-quality RCTs are still needed to confirm the role of colistin plus meropenem combination therapy in the treatment of drug-resistant *Acinetobacter baumannii* infection.

## Figures and Tables

**Figure 1 jcm-11-03239-f001:**
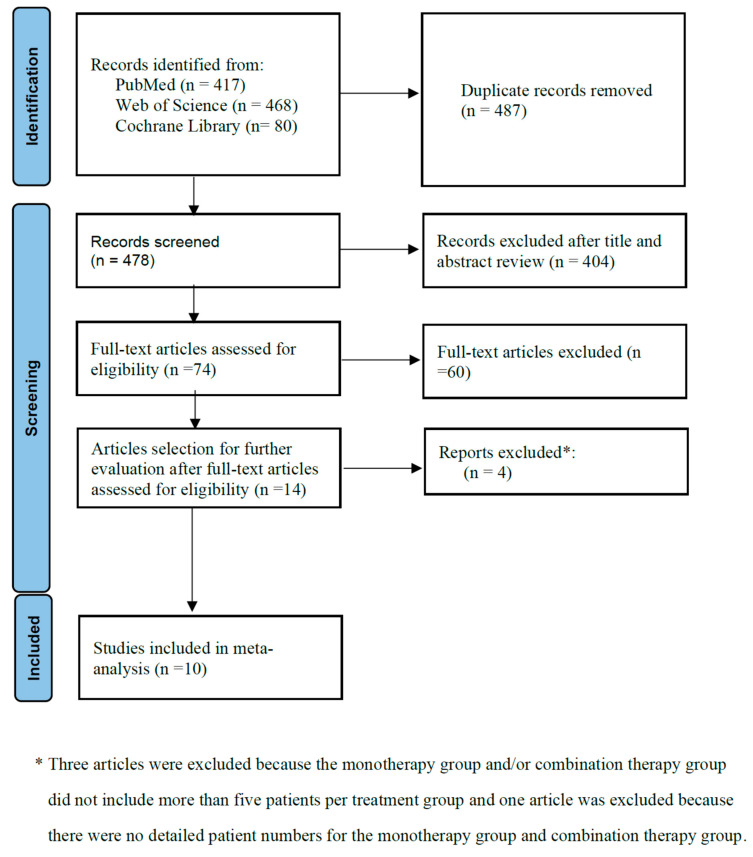
Flow diagram of the study process. Ten studies were included in the meta-analysis. Three were RCTs, and seven were retrospective observational studies.

**Figure 2 jcm-11-03239-f002:**
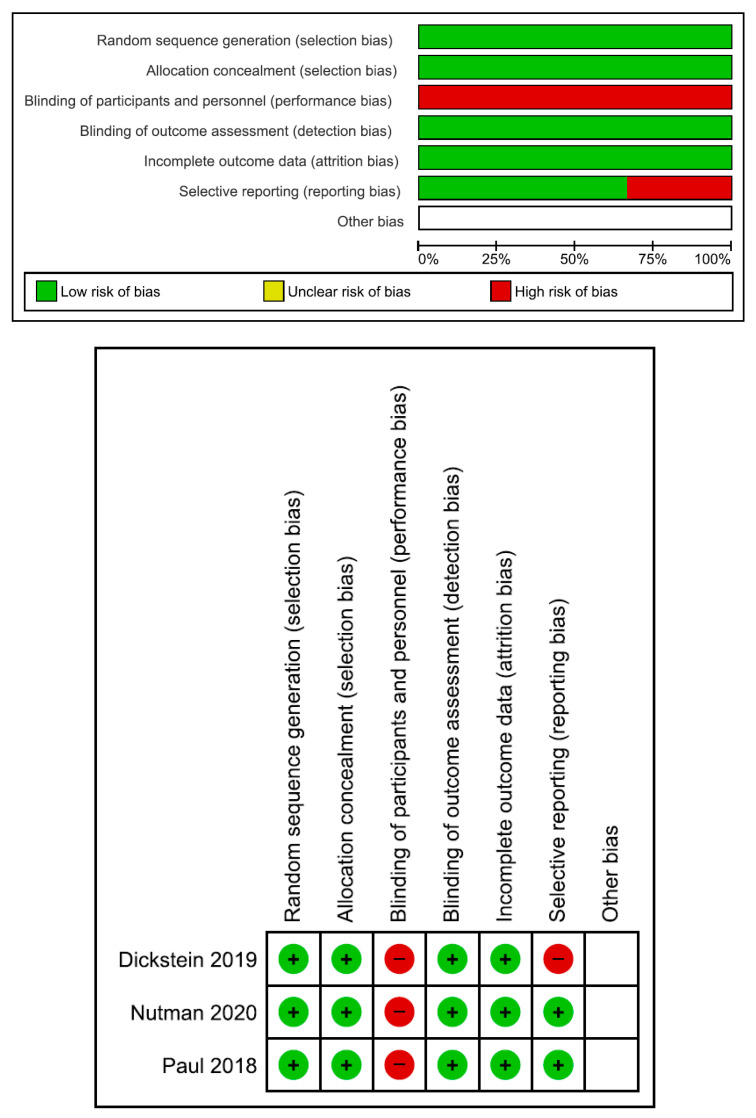
Risk of bias of three randomized controlled trials [26,29,30].

**Figure 3 jcm-11-03239-f003:**
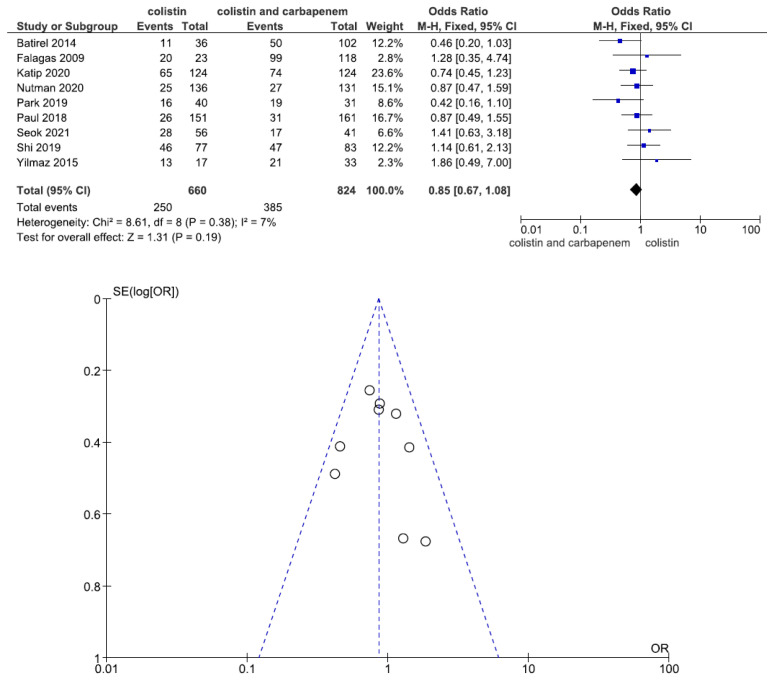
Forest plots and funnel plots for clinical improvement between colistin monotherapy and colistin plus meropenem combination therapy. Nine studies involving 1484 patients (660 with colistin monotherapy, 824 with colistin plus meropenem combination therapy) reported clinical improvement. There was no significant difference in clinical improvement between patients treated with colistin monotherapy and colistin plus meropenem combination therapy [23,24,25,26,27,28,30,31,32].

**Figure 4 jcm-11-03239-f004:**
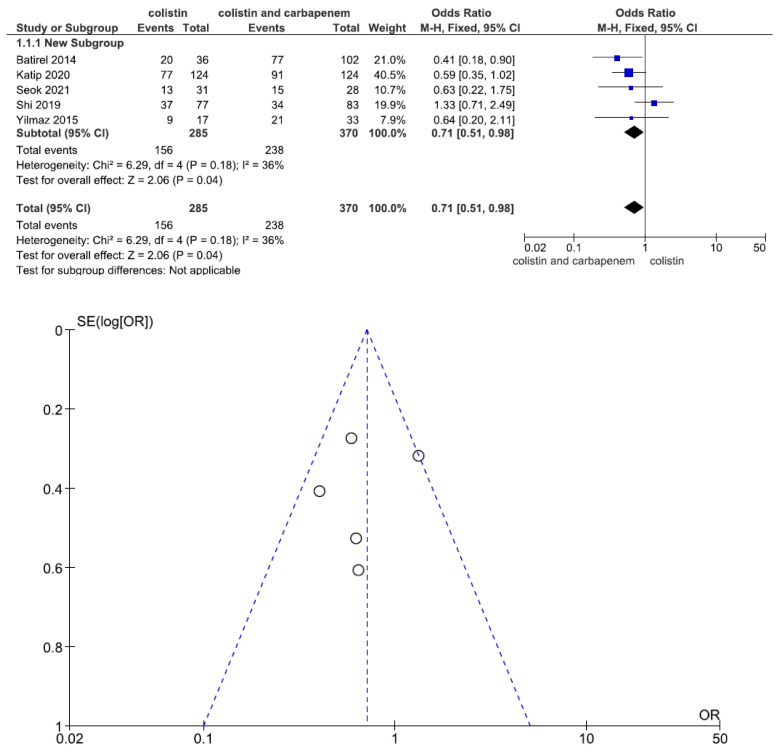
Forest plots and funnel plots for microbiological response between colistin monotherapy and colistin plus meropenem combination therapy. Five studies involving 655 patients (285 with colistin monotherapy, 370 with colistin plus meropenem combination therapy) reported the microbiological response. The overall microbiological response was significantly different between the two groups. The combination therapy patient group had a better microbiological response than the monotherapy patient group [24,25,28,31,32].

**Figure 5 jcm-11-03239-f005:**
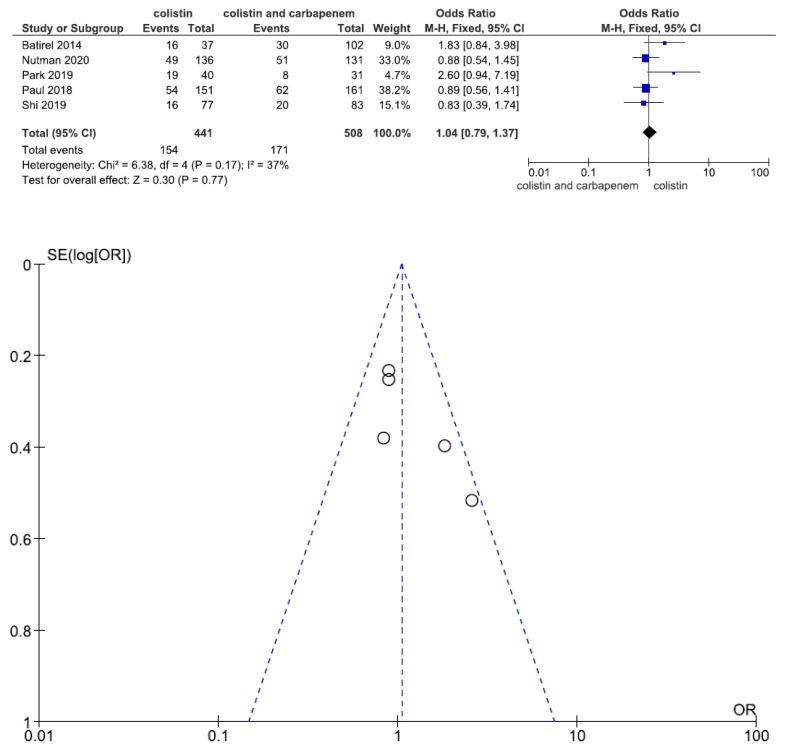
Forest plots and funnel plots for 14-day mortality between colistin monotherapy and colistin plus meropenem combination therapy. Five studies involving 949 patients (441 with colistin monotherapy, 508 with colistin plus meropenem combination therapy) reported 14-day mortality. There was no significant difference in 14-day mortality between patients treated with colistin monotherapy and colistin plus meropenem combination therapy [24,26,27,28,30].

**Figure 6 jcm-11-03239-f006:**
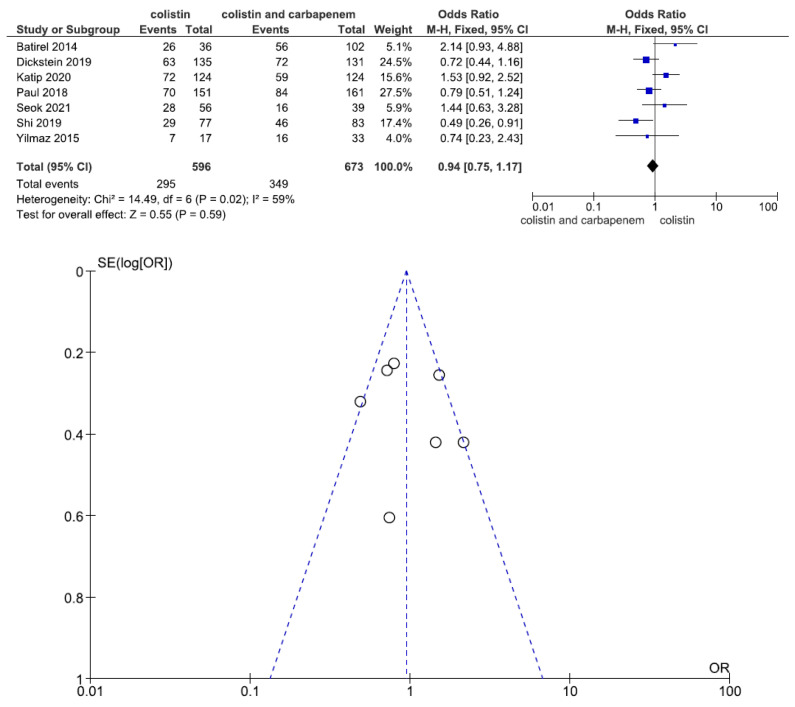
Forest plots and funnel plots for hospital mortality improvement between colistin monotherapy and colistin plus meropenem combination therapy. Seven studies involving 1269 patients (596 with colistin monotherapy, 673 with colistin plus meropenem combination therapy) reported hospital mortality or 28-day mortality. There was no significant difference in hospital mortality or 28-day mortality between patients treated with colistin monotherapy and colistin plus meropenem combination therapy [24,25,26,28,29,31,32].

**Figure 7 jcm-11-03239-f007:**
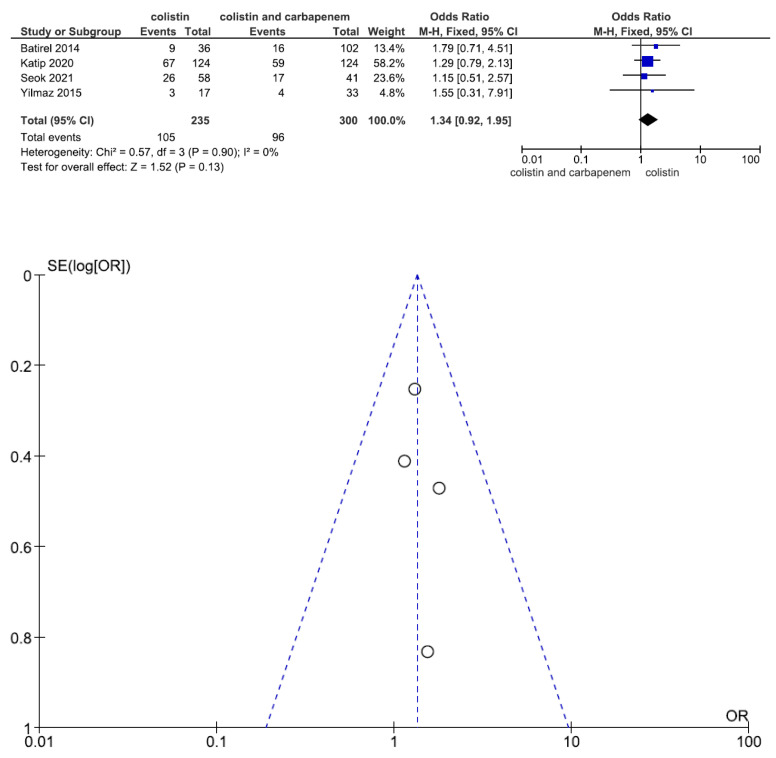
Forest plots and funnel plots for nephrotoxicity between colistin monotherapy and colistin plus meropenem combination therapy. Data regarding nephrotoxicity were reported in four studies including 535 patients (235 with colistin monotherapy, 300 with colistin plus meropenem combination therapy). There was no significant difference in nephrotoxicity between patients treated with colistin monotherapy and colistin plus meropenem combination therapy [24,25,31,32].

**Table 1 jcm-11-03239-t001:** Characteristics of the included studies.

Author/Year	Region	Study Type	Bacteria	Infection Type	No. of Patients (M)	No. of Patients (C)
Falagas, M.E., 2009 [23]	Greece	RETS	MDR AB (65.9%)	Mixed *	23	118
Batirel, A., 2014 [24]	Turkey	RETM	XRD AB	BSI	36	102
Yilmaz, G.R., 2015 [25]	Turkey	RETS	MDR AB XRD AB	VAP	17	33
Paul, M., 2018 [26]	Israel	RCTM	CR AB	BSI, VAP, HAP, UTI	151	161
Park, S.Y., 2019 [27]	Korea	RETS	CR AB	BSI	40	31
Shi, H., 2019 [28]	Korea	RETS	CR AB	Pneumonia	77	83
Dickstein, Y., 2019 [29]	Israel	RCTM	CR AB	BSI, VAP, HAP, UTI	135	131
Nutman, A., 2020 [30]	Israel	RCTM	CR AB	BSI, VAP, HAP, UTI	136	131
Katip, W., 2020 [31]	Thailand	RETS	CR AB	BSI, pneumonia, UTI, surgical site infection	124	124
Seok, H., 2021 [32]	Korea	RETM	CR AB	BSI, UTI, Pneumonia	58	41
**Author/Year**	**CST** **Loading Dose**	**CST** **Maintain Dose**	**Meropenem Maintain Dose**	**Duration of** **Treatment**		
Falagas, M.E., 2009 [23]	No data	No data	No data	17.9 days		
Batirel, A., 2014 [24]	No loading dose	5 mg/kg/day	1.0 gm q8h	No data		
Yilmaz, G.R., 2015 [25]	No data	4.5 MU q12 h	1.0 gm q8h	M:12.3 days C: 11.7 days		
Paul, M., 2018 [26]	9.0 MU	4.5 MU q12 h	2.0 gm q8h	No data		
Park, S.Y., 2019 [27]	No data	4.5 MU q12h	1.0 gm q8h	M: 8.88 days C: 8.22 days		
Shi, H., 2019 [28]	No data	4.5 MU q12h	2–3 gm/day	M: 12 days C: 14 days		
Dickstein, Y., 2019 [29]	No data	No data	No data	No data		
Nutman, A., 2020 [30]	9.0 MU	4.5 MU q12h	2.0 gm q8h	No data		
Katip, W., 2020 [31]	9.0 MU	4.5 MU q12h	1.0 gm q8h	No data		
Seok, H., 2021 [32]	No data	No data	No data	No data		

M: monotherapy; C: combination therapy; RETS: single center retrospective study; RETM: multicenter retrospective study; RCTM: multicenter randomized controlled trial; MDR: multidrug-resistance; XRD: extensive drug resistance; CR AB: carbapenem resistant *Acinetobacter baumannii*; BSI: blood stream infection; VAP: Ventilator-associated pneumonia; HAP: Hospital-acquired pneumonia; UTI: Urinary tract infection; CST: colistin; *: The infection site included pneumonia, urinary tract infection, skin/soft tissue infection, bacteremia, surgical infection, abdomen infection, orthopedic infection, catheter infection, and cerebrospinal fluid infection.

**Table 2 jcm-11-03239-t002:** Risk bias of seven retrospective cohort studies.

Author/Year	Confounding	Selection	Interventions Classification	Interventions Deviations	Missing Data	Measurement of Outcomes	Selective Results
Flagas, M.E., 2009 [23]	serious risk	serious risk	serious risk	serious risk	low risk	serious risk	serious risk
Batirel, A., 2014 [24]	serious risk	serious risk	serious risk	serious risk	low risk	serious risk	low risk
Yilmaz, G.R., 2015 [25]	serious risk	serious risk	serious risk	moderate risk	low risk	serious risk	low risk
Park, S.Y., 2019 [27]	serious risk	serious risk	serious risk	serious risk	low risk	serious risk	low risk
Shi, H., 2019 [28]	serious risk	serious risk	serious risk	low risk	low risk	serious risk	low risk
Katip, W., 2020 [31]	serious risk	moderate risk	low risk	low risk	low risk	moderate risk	low risk
Seok, H., 2021 [32]	serious risk	moderate risk	low risk	low risk	low risk	moderate risk	low risk

**Table 3 jcm-11-03239-t003:** Summary of meta-analyses of colistin monotherapy versus colistin combination therapy for the treatment of multidrug-resistant gram-negative bacteria infections.

Author/Pathogen	Clinical Improvement	Microbiological Response	14-Day Mortality	* Hospital Mortality	Nephro-Toxicity	Combination Antibiotics
Chen/AB	No	Yes	No data	No	No data	RIF, SUL, CAR, TGC UNA, CPZ, AMG, TZA MIN, TMP
Zusman, O.,/GNB1 [34]	No data	No data	No data	No	No data	CAR, TGC, RIF, AMG, SUL, VAN, TZA, FOS
Kengkla, K.,/AB [35]	No	Yes ^%^	No data	No	No	RIF, TGC, CAR, SUL, AMG, CPZ, UNA, FOS, MIN, TMP, TZA,
Vardakas, K.Z.,/GNB1 [36]	No data	No data	No data	No	No data	RIF, CAR, AMG, TGC, FOS, SUL, CIP
Cheng, I.L.,/GNB2 [37]	No data	No	No data	No	No	RIF, FOS, MPM, UNA
Wang, J.,/AB [38]	No	Yes	No data	No	No	CAR or/and Sul (subgroup analysis)
Schmid, A.,/AB [39]	No	No data	No data	No	No data	RIF, SUL, CAR, TGC UNA, AMG, TZA, RIF,
Samal, S.,/GNB1 [40]	No data	No data	No data	No	No data	RIF, VAN, TGC, CAR, SUL, AMG, CPZ, UNA, TAZ, FOS, MIN, TMP.

* Hospital mortality or 28-day mortality. ^%^ Colistin in combination with sulbactam was associated with a significantly higher microbiological response than colistin monotherapy, AB: *Acinetobacter baumannii*, GNB1: *Acinetobacter baumannii, Klebsiella pneumoniae,* and *Pseudomonas aeruginosa*, GNB2: *Acinetobacter baumannii, Pseudomonas aeruginosa,* and *Enterobacteriaceae.* MPM: meropenem, SUL: sulbactam, CAR: carbapenem, TGC: tigecycline, AMG: aminoglycosides, FOS: fosfomycin, RIF: rifampicin, CPZ: cefperazone/sulbactam, UNA: ampicillin/sulbactam: MIN: minocycline, TAZ: piperacillin/tazobactam, TMP:TMP/SMX, CIP: ciprofloxacin, VAN: Vancomycin.

## Data Availability

The datasets generated during and/or analyzed during the current study are not publicly available but are available from the corresponding author on reasonable request.

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
