# Peer review of "Colistin Monotherapy versus Colistin plus Meropenem Combination Therapy for the Treatment of Multidrug-Resistant Acinetobacter baumannii Infection: A Meta-Analysis"

_jcm, 2022, doi:10.3390/jcm11113239_

Round 1
Reviewer 1 Report
The study is overall fascinating and informative. The authors did lots of background studies and analyses to perform the remarks. But the study flow is a bit complicated for the familiar readers as well as early-stage researchers also. In my suggestion from sampling to methodology and analysis should be present is a clear workflow structure.
Reviewer 2 Report
Thank you for the opportunity to review this manuscript. The article presents a topic that is a major concern for researchers in the last decade. There are a lot of debates about the criteria that need to be taken into consideration in order to decide which is a better choice for the patient, mono/combined antimicrobial therapy, especially when the in-vitro results show a better antimicrobial activity for the combined therapy.
Overall, the manuscript is well structured, the results of the study are clearly described, and the conclusion supports the presented results.
However, I have a few comments for the authors, in order to improve the quality of the manuscript.
Minor concerns:
· The authors should use the same acronyms in the whole text (please see MDR AB/XDR AB, etc. in table 1 versus manuscript, where no abbreviation is described for MDR Acinetobacter baumannii
· In the abstract, keywords and a few places in the manuscript (line 223, 240) Acinetobacter baumannii should be formatted as italic characters
· Table 1 – RETS/RETM from the last two lines - the E seems to be written with different characters
· Table 2 – confounding, selection should be written with capital letters
· Figure 1 – the information about the location of the study may not be necessary in the legend, since it is described in the manuscript and is not directly related to the figure
Major concerns
· The introduction section should be reevaluated
o Line 72-80 contain a few sentences that are not very well related, seems to be just different information
o Line 72-75 – We?
o Information about the antimicrobial mechanism of colistin as mono versus combined therapy should be added
· Discussion section
o In the present form this section seems to be only a list of results from different studies
o The authors should present their analyze, opinion regarding their results when comparing them with the results from other studies.
Round 2
Reviewer 2 Report
Thank you for addressing my comments. The quality of the manuscript was improved.